# Seed Storage Physiology of *Lophomyrtus* and *Neomyrtus*, Two Threatened Myrtaceae Genera Endemic to New Zealand

**DOI:** 10.3390/plants12051067

**Published:** 2023-02-27

**Authors:** Karin van der Walt, Jayanthi Nadarajan

**Affiliations:** 1Ōtari Native Botanic Garden, Wellington City Council, 150 Wilton Road, Wellington 6012, New Zealand; 2School of Agriculture and Environment, Massey University, Palmerston North 4410, New Zealand; 3The New Zealand Institute for Plant and Food Research Limited, Fitzherbert Science Centre, Batchelar Road, Palmerston North 4474, New Zealand

**Keywords:** cryopreservation, desiccation, differential scanning calorimetry (DSC), lipid, myrtle rust, seed banking, seed physiology, seed storage behaviour, threatened

## Abstract

There is no published information on the seed germination or seed storage physiology of *Lophomyrtus bullata*, *Lophomyrtus obcordata*, and *Neomyrtus pedunculata.* This lack of information is hampering conservation efforts of these critically endangered species. This study investigated the seed morphology, seed germination requirements, and long-term seed storage methods for all three species. The impact of desiccation, desiccation and freezing, as well as desiccation plus storage at 5 °C, −18 °C, and −196 °C on seed viability (germination) and seedling vigour was assessed. Fatty acid profiles were compared between *L. obcordata* and *L. bullata*. Variability in storage behaviour between the three species was investigated through differential scanning calorimetry (DSC) by comparing thermal properties of lipids. *L. obcordata* seed were desiccation-tolerant and viability was retained when desiccated seed was stored for 24 months at 5 °C. *L. bullata* seed was both desiccation- and freezing-sensitive, while *N. pedunculata* was desiccation-sensitive. DSC analysis revealed that lipid crystallisation in *L. bullata* occurred between −18 °C and −49 °C and between −23 °C and −52 °C in *L. obcordata* and *N. pedunculata*. It is postulated that the metastable lipid phase, which coincides with the conventional seed banking temperature (i.e., storing seeds at −20 ± 4 °C and 15 ± 3% RH), could cause the seeds to age more rapidly through lipid peroxidation. Seeds of *L. bullata*, *L. obcordata* and *N. pedunculata* are best stored outside of their lipid metastable temperature ranges.

## 1. Introduction

Myrteae, a tribe in the Myrtaceae family, includes most of the species with fleshy fruits and, within New Zealand, this tribe is represented by two endemic genera, *Lophomyrtus* and *Neomyrtus* [1]. *Lophomyrtus* comprises two species: *L. bullata* (Soland. ex A. Cunn.) Burret and *L. obcordata* (Raoul) Burret [2], while *Neomyrtus* is represented by a single species, *N. pedunculata* (Hook.f.) Allan. *Lophomyrtus* species are highly susceptible to myrtle rust (*Austropuccinia psidii*) with fruit infection and a high degree of seedling death likely to result in localised extinction [3,4]. The impact of myrtle rust on *N. pedunculata* is not known as this species has not been included in susceptibility tests to date. Due to this predicted impact from myrtle rust, the threat status of all three species is now classified as nationally critical in New Zealand [5].

The urgency of ex situ conservation strategies to prevent the extinction of species susceptible to myrtle rust has been highlighted in Australia [6,7] and New Zealand [8,9]. Ex situ conservation methods include the maintenance of cultivated plants as living collections, tissue cultured plants as in vitro collections, conventional seed banks, and cryopreservation [10]. The most suitable ex situ conservation method will largely depend on the seed storage classification of the species. Seed-bearing species, in most cases, can be found along a continuum of storage behaviour, which includes: (a) orthodox (desiccation- and freezing-tolerant); (b) recalcitrant (desiccation-sensitive); or (c) intermediate (sensitivity to desiccation, freezing, or both) [11,12]. 

In addition to desiccation and freezing sensitivity, other variables influencing seed longevity can be complex and may relate to aspects such as embryo size, seed mass, seed coat, cellular composition, taxonomy, genotype, and the environment [13,14]. Of these variables, seed lipid composition, particularly the thermal stability of the lipids at the storage temperature, is a key predictor of intermediate seed storage physiology for seeds that contain high oil content [13,15,16,17,18]. Triacylglycerols (TAG) consist of trihydric alcohol glycerol esterified and usually comprise long-chain fatty acids (C14–C22), and may determine whether dry seeds are damaged when exposed to freezing temperatures [19]. Analysis of seed composition, in particular lipid content and their fatty acid compositions, can aid in developing the most appropriate storage protocol for the seed. In addition, thermal analysis using differential scanning calorimetry (DSC) can characterise the physical properties, such as crystallisation and melting temperatures of lipids and their stability. Using the DSC, lipid phase changes (solid to liquid and vice versa) and their relevant heat transitions are monitored and analysed as a function of time and temperature [20]. This enables the characterisation of seed lipid thermal fingerprints and their physical stability at stored temperatures [16]. The range (onset to end) of lipid crystallisation and melting temperature ranges can indicate the stability of the lipid phase at a particular temperature at which seeds are stored [21]. 

The aim of long-term seed storage is to retain high seed germination; hence, knowledge on seed germination, including dormancy breaking, is a crucial part of the seed banking process. It is estimated that up to 75% of species in temperate broadleaved evergreen forests have some form of dormancy [22]. Assessment of seed storage physiology can be hampered by the inability to germinate seed due to dormancy or lack of data on optimum germination conditions and, therefore, research into seed germination should be prioritised for these species [23]. Seed germination tests are the optimal way to monitor the physiological status of the seed; however, in some species, results from germination tests could be misleading due to ineffective germination conditions or partially effective dormancy breaking treatments [24]. For seed in storage, loss of fitness due to aging prior to seed mortality might also influence germination results [25]. 

Various aspects of seed physiology are particularly relevant to the long-term storage of seed. This includes an understanding of seed viability and dormancy, the ability of a seed to tolerate desiccation during seed development, seed ageing and how it affects viability loss, and how long seed lots will remain above viability thresholds [26]. These physiological aspects have been poorly studied for many native New Zealand species, including *Lophomyrtus* and *Neomyrtus* [27]. This lack of information is hampering effective ex situ conservation of Myrtaceae species threatened by myrtle rust. The aims of this study were to: (1) understand seed morphology and fruit and seed development for *L. bullata*, *L. obcordata*, and *N. pedunculata*; (2) investigate seed storage physiology by testing desiccation sensitivity, and the combined effects of desiccation and low temperature exposure and long-term storage of seed; and (3) to understand seed composition and lipid thermal profiles to predict seed storage behaviour, and to optimise long-term seed storage protocols for these species. 

## 2. Results

### 2.1. Fruit and Seed Morphology of L. bullata, L. obcordata, and N. pedunculata 

*L. bullata* fruit and seed were significantly larger compared with *L. obcordata* and *N. pedunculata*, and seed had higher moisture content (MC) at 0.52 g/g compared with *L. obcordata* (MC 0.35 g/g) and *N. pedunculata* (MC 0.15 g/g) (Figure 1; Table 1).

### 2.2. Biochemical Composition of Lophomyrtus Seed

Fresh *L. bullata* seed contained more total solids with higher fat and crude protein content compared with *L. obcordata* (Figure 2). Starch and sugar accounted for less than 1% of the seed composition in both species. The fatty acid profile showed that both *Lophomyrtus* species’ seed consisted mostly of polyunsaturated fatty acids (PUFA), specifically linoleic acid (Table 2). *L. bullata* seed stored for 15 months at 5 and −196 °C showed 0.8 and 1.0% decrease in saturated fatty acid (SAFA) content and 0.3 and 1.3% decrease in monounsaturated fatty acids (MUFA), whilst showing an increase of 1.2 and 2.3% in PUFA, respectively (Table 2). *L. obcordata* contained small amounts of undecamolic acid (C11), which was not present in *L. bullata*. 

### 2.3. Seed Germination Test and Seedling Vigour Assessment

*L. bullata* seed collected in April or May failed to germinate within six weeks. However, when seeds were collected in June, radicle emergence was recorded within 10 days, and maximum germination was reached within six weeks. Radicle emergence was significantly higher in fresh non-stratified seed (100%) compared with seed stratified dry for four weeks (74%). Germination and seedling height were similar in non-stratified (46.9% and 47.1 mm) and stratified (45.9% and 45.3 mm) seed (Table 3). Stratification did not affect radicle emergence or germination in *L. obcordata* with final germination of 85% and 82.5% for non-stratified and stratified seed, respectively. Seedling height ranged from 12.3 to 35 mm with the mean seedling height after eight weeks 35.7 mm and 34.5 mm for non-stratified and stratified seed, respectively (Table 3). Seed collected from natural *L. bullata x L. obcordata* hybrids did not have dormancy with radicle emergence similar in non-stratified (82%) and stratified (78%) seed. Germination was lower than radicle emergence for stratified and non-stratified seed (71% and 66%, respectively) (Table 3). Parameters characterising seedling vigour was not recorded in the natural hybrid, *L. bullata x L. obcordata*. 

Stratification time (4 vs. 20 weeks) and method (dry vs. wet) influenced radicle emergence in *N. pedunculata* seed. Dry stratification failed to stimulate radicle emergence irrespective of the stratification period (Table 4). Wet stratification for 4 weeks stimulated radicle emergence (38%) with significantly higher radicle emergence after 20 weeks of wet stratification (72%; *p* = 0.014; Table 4). All the seeds that produced a radicle completed the germination process (Table 4). Seeds exposed to wet stratification for up to 20 weeks started to germinate during stratification (Figure 3A), while those exposed to dry stratification showed no visual signs of germination during the stratification period (Figure 3B). Wet stratification time (4 vs. 20 weeks) did not have a significant impact on seedling height; although, seedlings regenerated from seed stratified wet for 20 weeks were slightly taller (20 mm) compared with those stratified for 4 weeks (15 mm). 

### 2.4. Seed Desiccation Trial

*L. bullata* seeds displayed desiccation sensitivity with a decrease in radicle emergence corresponding to a decline in MC (Table 5). Desiccation to 30% equilibrated relative humidity (eRH) resulted in a decrease in viability; although this was not significant. Further desiccation to 15% eRH decreased radicle emergence and germination to 48% and 39%, respectively, with seedling height decreasing from 47 mm in fresh seed to 31 mm following desiccation (Table 5). Although germination declined and seedling vigour was reduced when MC was lowered, this difference was not statistically significant. Desiccation, irrespective of the eRH, had no impact on *L. obcordata* seed viability or seedling vigour. Radicle emergence in *L. bullata x L. obcordata* was reduced from 82% in non-desiccated seed to 55% in seed desiccated to 15% eRH with all seed showing radicle emergence completing the germination process (Table 5). *N. pedunculata* seed desiccated to 15% eRH had low (4%) radicle emergence and germination compared with 72% in non-desiccated seeds. The seedlings of desiccated seed also only reached 50% of the height recorded by seedlings from non-desiccated seeds (Table 5).

### 2.5. Combination of Desiccation and Freezing Sensitivity Assessment

*L. bullata* displayed freezing sensitivity at −18 °C and −196 °C with a decline in radicle emergence, germination, and seedling vigour associated with the lower storage temperature. This decline was accelerated by lowering the MC. Interestingly, for seed desiccated to 15% eRH, germination and seedling vigour remained constant with similar results for the non-stored seeds for both 5 °C and −18 °C. However, germination and seedling vigour declined significantly in seed exposed to −196 °C (Table 6). *L. obcordata* showed an overall declining trend in radicle emergence, germination, and seedling vigour following freezing at −18 °C and −196 °C; although the decline in most cases was not statistically significant (Table 6). The exception was seed desiccated to 15% eRH and exposed to −196 °C, which retained radicle emergence and germination (84% and 74%, respectively), which was comparable to fresh seed (85%). Similarly, there was an overall reduction in seedling vigour (height and number of leaves) following freezing (Table 6) except for seed desiccated to 30% eRH and exposed to −18 °C. *L. bullata x L. obcordata* displayed sensitivity to storage at −18 °C with a non-significant decline in radicle emergence (33%) and a significant decrease in germination (29%) compared with seed exposed to 5 °C and −196 °C (Table 6). Parameters associated with seedling vigour were not recorded in *L. bullata x L. obcordata*. For *L. bullata*, *L. obcordata*, and *L. bullata x L. obcordata*, the interactive effect between desiccation and temperature on radicle emergence was not significant. Desiccation was found to be the most influential explanatory value. *N. pedunculata* showed a decline in viability following exposure to −18 °C (Table 6). Since seed germination was limited due to dormancy, no seedling vigour assessment was carried out.

### 2.6. Seed Storage Trial

*L. bullata* seed stored for 24 to 36 months showed extreme sensitivity to freezing at −18 °C and −196 °C with a significant decline observed in radicle emergence and germination. Desiccation was, however, essential to retain viability in seed stored at 5 °C (Table 7). Time in storage, seed lot, or the combined effect of time and seed lot, did not have a significant influence on the viability of seed in storage, with temperature found to be most influential variable. Overall seedling vigour was not affected by storage temperature, except for the single seedling obtained from seed stored at −18 °C, which was significantly taller (40 mm) and produced more leaves (2.0) compared with other treatments (Table 7). The *L. obcordata* seed storage trial was represented by three seed lots: Ōtari 19, Matai, and Old Mill. Two of the seed lots, Ōtari 19 and Matai, showed freezing sensitivity, illustrated in the decline in viability, radicle emergence, germination, and seedling height (Table 7). Response to storage at −18 °C was also influenced by seed lot, with seed from Old Mill having significantly higher radicle emergence, germination, and seedling height compared with Ōtari 19 and Matai. The interactive effect of time in storage, temperature, and seed lot was not significant. Seed from *L. bullata x L. obcordata* also continued to decline in viability after 12 months at sub-zero temperature storage (Table 7). Interestingly, although radicle emergence was lower in seed stored at −18 °C (42%) and −196 °C (33%) compared with seed stored at 5 °C (55%), the difference was not significant (*p* = 0.170). In contrast to radicle emergence, germination was significantly higher in seed stored at 5 °C (*p* = 0.001; 42.5%) compared with germination in seed stored for 12–36 months at −18 °C (19.3%) and −196 °C (22%) (Table 7). Seedling vigour was similar irrespective of the storage temperatures. 

### 2.7. Differential Scanning Calorimetry

The DSC thermograms provided measurements for critical parameters including enthalpy of phase transitions that occur between −90 °C and 20 °C. For the cooling cycle, all three species showed two lipid crystallisation events with onset and end temperatures spread from −18 °C to −50 °C (Figure. 4). The onset temperature for lipid crystallisation in *L. bullata* is slightly higher, i.e., −18 °C compared with −23 °C and −25 °C in *L. obcordata* and *N. pedunculata*, respectively (Figure 4). There was a small ice crystallisation event noted for *L. bullata* seeds equilibrated to 30% eRH with seed moisture content c. 0.17 g/g (Figure 4A). 

Upon warming, all three species displayed three lipid melting events spread from −77.4 to 5.7 °C (Appendix A). The graph for *L. bullata* seed equilibrated to 30% eRH with seed MC of 0.17 g/g had a fourth peak, which is related to ice melt (Figure 5A). The lipid melting events for *L. bullata* seeds ranged from −19.7 to 4.8 °C, whereas for *L. obcordata* it ranged from −24.3 to 3.7 °C and −25.3 to 1.2 °C for *N. pedunculata*. Melting events spanned similar temperature ranges (24 to 28 °C) in all three species. The average of total melt enthalpy (cumulative value of all three melting events) was 3.1 J/g for *L. bullata* and 4.3 J/g for *L. obcordata*, with the highest enthalpy recorded in *N. pedunculata* at 6.6 J/g (Appendix A). 

## 3. Discussion

This study investigated the seed biology and storage physiology of *Lophomyrtus* and *Neomyrtus*, two threatened genera of Myrtaceae endemic to New Zealand, in order to recommend optimum storage conditions for the long-term conservation of these species. 

Mature *L. bullata* seed had a significantly higher MC (0.52 g/g) compared with *L. obcordata* (0.35 g/g) (Table 2). Desiccation-sensitive seeds from rainforest species in Australia and Africa all had moisture contents at maturity of greater than 0.5 g/g (~35% fresh weight) [28,29]. Seed moisture content of <25% could be used as a predictor of desiccation tolerance [30]. Another important seed characteristic for seed storage is lipid content, particularly the physical phase of the lipid at storage temperatures [15,28]. Seed maturity is often associated with the accumulation of specific biochemical compounds related to the seed’s ability to tolerate desiccation and survive prolonged periods of unsuitable conditions prior to germination [31]. Loss of germination in stored seed is accompanied by a decrease in embryo viability and deterioration of storage nutrients [32]. Since the degradation of storage lipids can have important physiological and ecological consequences for seed, an understanding of the seed’s fatty acid composition can aid in the optimisation of seed storage protocols. During storage, fatty acids are vulnerable to lipid degradation through oxidation or enzymatic hydrolysis [33]. Similarly, Hamilton et al. [28] found that several characteristics, including seed lipid content, could be used to flag Australian rainforest species that have unknown seed storage behaviour. The total lipid content of *L. bullata* and *L. obcordata* seed in this study was 7.6% and 4.1% of the total seed weight, respectively (Figure 4). Interestingly, linoleic acid (C18:2) made up 75% of the fatty acids in both species.

Linoleic acid is abundant in many species, including *Vitis vinifera* (grape), *Nicotiana tabacum* (tobacco), and *Ribes nigrum* (blackcurrant) [34]. Polyunsaturated fatty acids (PUFAs), which includes linoleic acid, are particularly prone to lipid oxidation during the seed aging process [32]. This study found that there was a slight increase in linoleic acid from 75% in fresh *L. bullata* seed to 77% and 78% in desiccated *L. bullata* seed stored for 15 months at 5 °C and −196 °C, respectively (Table 2). Most studies report a decrease in linoleic acid in aged seed, for example, almond seed [35], various rice species [36], tomatillos [37], and orchids [34]. It is thus expected that lipid content and the total amount of fatty acids in ageing seeds would decrease due to lipid oxidation [32]. The changes in fatty acid composition reported in this study could indicate that the molecular structure of the crystals are reorganising into denser and lower energy forms, as suggested by Mira et al. [16]. However, since these changes are relatively small, it is postulated that in *L. bullata* seed, particularly the increase in linoleic acid is likely due to intraspecific variation in the composition of the seed collected at different development stages and because of different genotypes. 

Seed germination tests are considered the ultimate method for assessing seed viability. This, however, requires knowledge of seed dormancy, and the identification of optimum germination conditions. Species occurring in climates with seasonal variation (summer and winter) often have seed that becomes non-dormant over winter, resulting in germination during spring [22]. Winter rainfall in these areas furthermore increases the likelihood that wet cold stratification will be required to break dormancy. *N. pedunculata* seed in this study was collected from Taranaki, which is in the central North Island of New Zealand. It is postulated that high winter rainfall, coupled with low minimum temperatures during winter, breaks dormancy and facilitates germination in spring. Dormancy that requires a longer cold stratification period is likely a strategy to prevent early emergence of seedlings during the cold winter months [38]. Dormancy can furthermore vary between populations and genotypes. For example, Anderson and Milberg [39] found that seed dormancy varied between mother plants, populations, and years in *Silene noctiflora*, *Sinapis arvensis*, and *Spergula arvensis*. Similarly, dormancy breaking requirements varied between three *Actinidia* species; although, all the species showed increased germination after five weeks’ cold stratification [40]. It is recommended that *N. pedunculata* seed is collected from different populations and tested to establish if dormancy is present. 

Conventional seed banking requires desiccation to about 15% eRH and the storage of seeds at −20 ± 4 °C [41,42]. Desiccation tolerance in *L. bullata* and *L. obcordata* was investigated by drying the seeds to 30% and 15% eRH. Neither species suffered a significant decrease in radicle emergence or germination following desiccation to 30% eRH. Further desiccation to 15% eRH did not have an impact on *L. obcordata*, while a significant decrease in radicle emergence and germination was recorded in *L. bullata* (Table 5). The desiccation sensitivity in *L. bullata* in this study contrasted with that reported by Nadarajan et al. [8] where *L. bullata* seed was found to be desiccation-tolerant. The standard deviation observed for radicle emergence and germination in *L. bullata* in this study was high (21 and 17%, respectively). This could indicate that there is intraspecific diversity towards desiccation tolerance (15% eRH) [43]. Intraspecific desiccation tolerance can be variable and influenced by factors such as seed development at time of harvesting, parental plant characteristics, post-harvest seed handling, and climatic conditions between years [11,44]. It was interesting to note that *L. bullata x L. obcordata* appears to share desiccation characteristics from both parent species with seed slightly less sensitive to desiccation compared with *L. bullata* (Table 5). This has also been found in cultivated species of *Coffea arabica*, which showed a level of desiccation tolerance intermediate between its two parental species [43]. The results from our study show that seed from *L. bullata*, *L. obcordata*, and *L. bullata x L. obcordata* falls along a continuum of desiccation tolerance, supporting the hypothesis that non-orthodox species display very high variation in desiccation sensitivity [43,45].

Seed with intermediate storage physiology can display tolerance to desiccation, but sensitivity to freezing [12,23]. To test for freezing sensitivity, *L. bullata* and *L. obcordata* seed desiccated to 30% eRH and 15% eRH were exposed to 5 °C, −18 °C, and −196 °C for 14 days. Although *L. bullata* seed tolerated desiccation to 30% eRH, freezing seed at this MC resulted in a decrease in radicle emergence and germination. This is likely due to the presence of freezable water, which resulted in the formation of ice crystals. Although there was an initial decrease in viability following desiccation to 15% eRH, viability remained similar when seed was stored at 5 °C and −18 °C, but a further decrease in viability was associated with storage at −196 °C. *L. bullata* seed stored for 24 to 36 months at −18 °C suffered significant viability loss. Seed from the same seed lot stored for 24 months at 5 °C did not lose significant viability compared with desiccated seed without storage. Since the seed from the same seed lot that was stored at 5 °C had greater viability than seeds stored at −18 °C, it can be concluded that *L. bullata* seeds do not tolerate storage at freezing temperatures.

*L. obcordata* seed was not sensitive to desiccation and although there was a decrease in seed viability following exposure to −18 °C for 14 days, it was not significant. However, when desiccated (15% eRH) seed was stored for 24 to 36 months at −18 °C, two collections (Matai and Ōtari 19) lost significant viability, while seed collected from Old Mill Road retained 63% radicle emergence (Table 4). Loss of seed viability during storage can be seed lot-specific [11,46], with some accessions rapidly losing viability in storage, while freezing-induced viability loss can take multiple years in others [47]. Intraspecific variability in seed physiology has been reported in species such as *Aesculus hippocastanum*, *Silene noctiflora*, *Sinapis arvensis*, *Spergula arvensis*, and *Thlaspi arvense* [39,48]. Although not fully understood, reasons for this intraspecific variability include genetic heterogeneity, climatic conditions, seed development stage, and post-harvest treatments [11,25,44]. The *L. obcordata* seed from Old Mill Road and Matai was collected at the same time and from the same general area (Marlborough in the South Island of New Zealand). It is, therefore, postulated that the difference in seed storage physiology reported here is likely due to genetic variation.

DSC was used to characterise the phase behaviour of lipids. The lack of large endothermic peaks suggest that no ice nucleation was present, except for *L. bullata* seeds that had been dried to 0.17 g/g (~30% eRH). Transitions observed in the cooling and warming thermograms are, therefore, due to lipids. The crystallisation and melt fingerprints of *L. bullata* seed spanned 87 °C (−71 °C (onset) to 16 °C (end)). Oily seeds that stored relatively poorly at −20 °C tend to have lipids with crystallisation and melting temperatures that spread over a wide range and spanned the storage temperature [16]. This study furthermore highlighted that when lipids were in a mixture of solid and liquid phase (metastable) at the storage temperature, the Brassicaceae seeds aged more rapidly. The DSC thermograms in our study indicated that lipid crystallisation of *L. bullata* occurred between −18 and −28 °C, thus overlapping the conventional seed banking temperature of −20 °C. Long-term storage of this seed at temperatures close to or within the lipid phase transitions should be avoided as this could lead to rapid seed deterioration due to unforeseen consequences to the seed cell structure [28,49]. For *L. obcordata* and *N. pedunculata*, the onset temperature of crystallisation is below −23 °C. Although slightly outside of the storage temperature, it is likely that the lipids are in a meta-stable state that could accelerate seed aging. In addition, it has been suggested that long-term storage of seed at specific temperatures could affect the lipid phase behaviour, especially onset and end of melt temperatures, enthalpies, and crystallisation events [16,50]. Interestingly, seed viability in *L. bullata*, *L. obcordata*, and *L. bullata x L. obcordata* was also not retained when desiccated seed was stored at −196 °C. At least two possibilities can be proposed to explain the low survival rate following exposure to liquid nitrogen (LN): (1) seeds were already aging prior to storage at −196 °C; or (2) the cooling and thawing conditions employed in this study were sub-optimal. The first possibility, in which seeds continued to age, has been highlighted by Walters [19], indicating that aging occurs in all seed whether in soil seed banks, conventional seed banks, or in cryostorage. However, in our study, seed viability remained constant in *L. bullata* and *L. obcordata* after storage for 24 months at 5 °C, while significant viability was lost when the same seed lots were stored at −18 °C. This was similar for *L. bullata x L. obcordata* seed stored for 12 months at 5 °C, −18 °C, and −196 °C. This indicates that over the course of the study period, freeze sensitivity, and not seed aging, caused viability loss in *L. bullata*, *L. obcordata*, and *L. bullata x L. obcordata*. 

The second possibility, which postulates that cooling and warming rates employed in the study were sub-optimal, is derived from studies by Vertucci [51] and Dussert et al. [17]. Sensitivity to rapid cooling of 100 °C to 200 °C min^−1^ has been reported in the dry seed of various *Coffea* spp. [17]. Slow cooling at 1 °C min^−1^ to −50 °C resulted in 70% germination in *Coffea arabica*, while direct submersion in LN was fatal [52]. Similarly, slow cooling of 3 °C h^−1^ was required for the cryopreservation of lettuce seed [53]. Our study was limited to rapid cooling through direct submersion of seed into LN, and slow cooling rates to minimise lipid crystallisation should be investigated. 

## 4. Materials and Methods

### 4.1. Plant Material

*L. bullata*, *L. obcordata*, and *L. bullata x L. obcordata* fruit was collected over the study period between April and June 2018–2021 from various localities in the lower North and upper South Island, while *N. pedunculata* fruit was collected from Taranaki in 2021 (Appendix A). Fruits were transported to the Lions Ōtari Plant Conservation Laboratory in Wellington where they were cleaned and stored at 5 °C until used. Seeds used for experiments that did not require stratifications were not stored prior to experiments.

### 4.2. Fruit and Seed Morphology of L. bullata, L. obcordata, and N. pedunculata

Randomly selected fruit from *L. bullata*, *L. obcordata*, and *N. pedunculata* were used to determine fruit weight (g), number of seed per fruit, seed weight (mg), seed size (mm), seed moisture content (expressed as g H_2_O/g seed dry weight), and seed germination (procedures described below). Fifty fruits per species were used to determine the number of seed per fruit, while seed weight was determined as the average weight of 10 seeds in 4 replicates weighed on a precision scale (Model: AUW120D, Shimadzu, Japan). To determine seed diameter, digital images were taken of 20 randomly selected seeds per species and measured using LC Micro 2.2 image analysis software Version 2.2 (Olympus). 

### 4.3. Biochemical Composition of Lophomyrtus Seed

Since information on seed biochemical composition, particularly fatty acids, can aid in developing an appropriate storage protocol for seed, this was analysed for fresh *L. bullata* and *L. obcordata* seed. In addition, fatty acid composition was also analysed for desiccated (~15% eRH) *L. bullata* seed stored for 15 months at 5 °C and −196 °C. Total solids, crude protein, fat, starch, sugars, and fatty acid profiles were analysed for fresh *L. bullata* and *L. obcordata* seed using standard methods of analysis prescribed by AOAC International [54]. The total fat and fatty acid composition was based on Sukhija et al. [55], with in-house modifications and included extraction, purification, and esterification, followed by gas chromatographic analysis. For fatty acid profiles, 2 g of ground seed was placed in a 50 mL beaker containing 2 mL benzene. After adding 10 mL HCl, the solution was incubated at 70–80 °C for 40 min. A total of 10 mL anhydrous methanol was then added to the mixture and, once the solution had cooled, it was poured together with 25 mL ether into the extraction tube of a Mojonnier fat-extractor. Following 1 min of mixing, an additional 25 mL of redistilled petroleum ether was added, and the solution was centrifuged for 20 min at 600 rpm. The ether–fat solution was filtered through a cotton pledget packed before re-extracting 15 mL of ether. After shaking the solution, clear ether was drawn and evaporated slowly before placing the dry fat in an oven at 100 °C to dry to a constant weight. Samples were then analysed using gas chromatography (GC2010; Shimidzu, Kyoto, Japan). 

### 4.4. Moisture Content Determination

MC for all treatments (fresh and desiccated seeds) were determined gravimetrically after drying at 103 ± 1 °C for 17 h [56]. Data presented represent the mean MC of ten seeds replicated four times and expressed as g H_2_O per g dry matter (g/g). 

### 4.5. Seed Germination Test and Seedling Vigour Assessment

Germination trials were conducted in 500 mL Magenta^®^ tissue culture vessels containing 100 mL of soil. Seeds were not surface sterilised before germination experiments. The presence of dormancy and the role of cold stratification (5 °C) on germination was determined. *L. bullata*, *L. obcordata*, and *L. bullata x L. obcordata* seed were planted for germination fresh on the day of collection and following dry stratification for four weeks at 5 °C. In our pilot study, fresh seed of *N. pedunculata* did not germinate within four weeks (data not shown) indicating the likely presence of dormancy [22]. Based on this, *N. pedunculata* seeds were either stratified dry (cleaned seed in glass vials) or moist (in fruit pulp) at 5 °C for 4 and 20 weeks before germination tests (Appendix A). All germination experiments were conducted in an incubator set at alternating temperatures of 15/25 °C with a 18 h photoperiod at a photosynthetic flux density of 30–50 μmol s^−1^ m^−2^ provided by cool-white fluorescent tubes. These temperatures were selected based on pilot studies which compared germination rates between 10/20 °C and 15/25 °C (data not shown). Radicle protrusion of at least 5 mm was the criterion for radicle emergence, with germination defined as the opening of cotyledonary leaves. Germination was assessed weekly until no new germination was observed for at least 2 weeks (6 weeks for *Lophomyrtus* species and 8 weeks for *N. pedunculata*). At the end of the germination period, non-germinated seeds were dissected to determine the presence of an embryo, as illustrated in Figure 1. Seedling height (mm) was measured from the end of the radicle to the top of the leaves, and the number of leaves (number of fully developed leaves) were recorded for all germinated seed. Due to limited seed quantities, each treatment consisted of a minimum of 10 seeds in 4 replicates. 

### 4.6. Seed Desiccation Trial

*L. bullata* and *L. obcordata*, seeds were desiccated to 15% and 30% eRH in airtight chambers kept at room temperature until they had fully equilibrated. These RH environments were created using lithium chloride (Sigma Aldrich, Auckland, New Zealand) salt solutions with different concentrations and are commonly applied in seed banking [57]. Seeds were weighed every second day until there was no more change in the weight. Unchanged weights indicated that seed moisture contents had equilibrated to the ambient RH. MC tests were carried out to verify seed moisture prior to further treatment. To prevent imbibition damage, desiccated seeds were rehydrated in a 100% humidity chamber for 24 h before germination tests. Due to a limited number of seed in two seed lots for *L. bullata* (Butchers and Wrights) and *L. obcordata* (Matai and Old Mill), viability testing was not conducted following desiccation. Furthermore, due to limited seeds in *L. bullata x L. obcordata* and *N. pedunculata*, these seeds were desiccated to 15% eRH only. 

### 4.7. Combination of Desiccation and Freezing Assessment

The combination of seed sensitivity to desiccation as well as low and sub-zero temperatures was evaluated on *L. bullata* (Kap 21) and *L. obcordata* (Ōtari 21) by exposing fresh and desiccated seeds (15% and 30% eRH) to 5 °C (fridge), −18 °C (freezer), and −196 °C (liquid nitrogen) for 14 days. Due to the limited number of seed, *N. pedunculata* seeds were desiccated to 15% eRH and exposed to −18 °C for 14 days. Viability was assessed through germination, as described above. 

### 4.8. Seed Storage Trial

The long-term storability was also evaluated for non-desiccated and desiccated seed (15% eRH) stored at 5 °C (fridge), −18 °C (freezer), and −196 °C (liquid nitrogen) for different collections of *L. bullata*, *L. obcordata*, and *L. bullata x L. obcordata*. The time in storage ranged from 12 to 36 months (Appendix A). 

### 4.9. Differential Scanning Calorimetry

Thermal behaviour of lipids in seed desiccated to 5% eRH, 15% eRH, and 30% eRH was determined using a PerkinElmer differential scanning calorimeter (DSC 8500) (Shelton, CT, USA), calibrated for temperature and heat flow with zinc (melting point 419.5 °C) and indium (melting point 156.6 °C). Seeds (3 seeds per treatment) of *L. bullata*, *L. obcordata*, and *N. pedunculata* were hermetically sealed into a large volume (60 µL) stainless steel capsule (sample pan) using an O-ring with the aid of a PerkinElmer universal crimper (Shelton, CT, USA). Sample weight was measured using a micro-balance (Model: XPR6UD5; Mettler-Toledo, Greifensee, Switzerland), and the sample was subjected to calorimetric assessment within 5 min of preparation. Samples were cooled from 25 °C to −100 °C and held for 1 min before rewarming to 25 °C at a rate of 10 °C min^−1^. The exothermic and endothermic heat changes derived from the crystallisation and melt endotherm during the cooling and warming cycles were regarded as lipid transitions. The onset temperature of the transition was determined as the temperature at which the tangent of the sharpest portion of the first peak intersected the baseline. The onset, peak, and end temperatures were calculated using PYRIS software (version 13.2). The areas of melt and crystallisation peaks were calculated using the area above the baseline and expressed as millijoule (mJ). The enthalpies for these transitions are presented as joule per gram of sample weight (J/g).

### 4.10. Statistical Analysis

All germination experiments involved a minimum of 10 seeds for each treatment, repeated 4 times. Germination results and seedling heights were compared using two-way ANOVA, followed by Fisher’s protected least significant difference (LSD) test for significantly different means (*p* < 0.05). A two-way ANOVA was used to investigate the relationship between desiccation and storage temperature on radicle emergence, germination, and seedling vigour (height and number of leaves). Arcsine square root transformation was used to normalise data where needed. ANOVA followed by Tukey’s honest significant difference (HSD) test was used to investigate differences in onset of melt temperature, freeze temperature, and enthalpy of the melt. All statistical analyses were performed at the 0.05 level of significance and results are expressed as mean ± SD. Statistical analysis was conducted using XLStat Software version 1.3 (2021) and SAS/STAT version 14.2.

## 5. Conclusions

The study revealed that seed of *L. bullata*, *L. obcordata*, *L. bullata x L. obcordata*, and *N. pedunculata* should be considered as intermediate in their storage physiology. *L. bullata* was sensitive to desiccation to MCs suitable for conventional seed banking (~15% eRH). Although seed viability and seedling vigour were improved following desiccation to 30% eRH, DSC analysis revealed ice nucleation due to the presence of freezable water. In addition, *L. bullata* was also freezing-sensitive, with seed stored at −18 °C affected by lipid crystallisation. *L. obcordata* seed was not desiccation-sensitive, but lost significant viability during sub-zero temperature storage. One seed lot was found to retain viability in seed stored at −18 °C, which could indicate intraspecific variation in storage physiology. Lipid thermal fingerprints of *L. obcordata* indicated that lipids are in a metastable state at conventional storage temperatures (i.e., −18 °C) and likely to continue deteriorating. No long-term data were available for *N. pedunculata*; although, this study revealed the species is desiccation-sensitive and likely to be freezing-sensitive with similar lipid thermal fingerprints to *L. obcordata.* It is recommended that seed of all three species are stored outside of lipid transition temperatures and that cooling rates are optimised. In addition, cryopreservation protocols using cryoprotectants should be investigated for *L. bullata*, which is confirmed as a desiccation-sensitive seed. 

## Figures and Tables

**Figure 1 plants-12-01067-f001:**
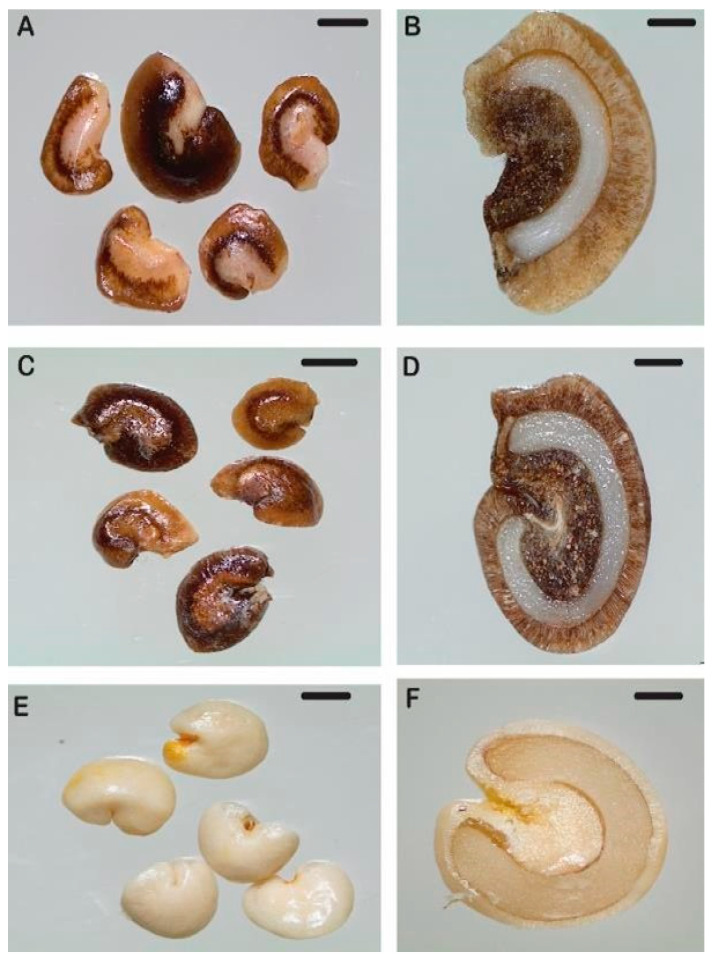
Whole and dissected seed illustrating embryo size and structure in *Lophomyrtus bullata* (**A**,**B**), *L. obcordata* (**C**,**D**), and *Neomyrtus pedunculata* (**E**,**F**). Scale bar = 1 mm.

**Figure 2 plants-12-01067-f002:**
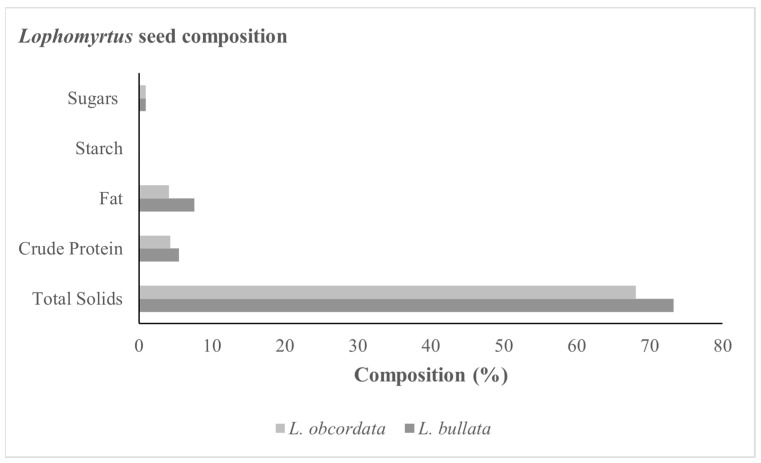
Biochemical composition of fresh *Lophomyrtus bullata* and *L. obcordata* seeds.

**Figure 3 plants-12-01067-f003:**
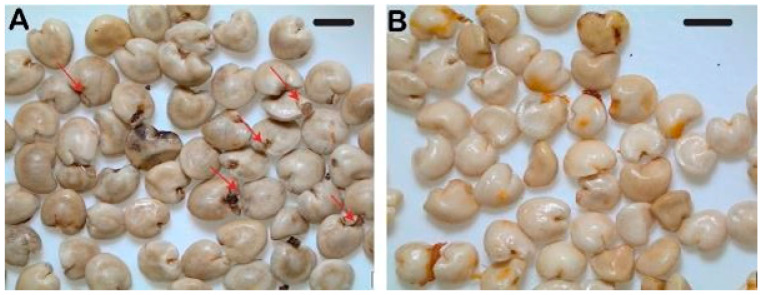
Radicle emergence was recorded in *Neomyrtus pedunculata* seeds during wet stratification (**A**) (red arrows indicate emerging radicles). Seed stratified in dry conditions did not display any radicle emergence during the stratification period (**B**). Black scale bars are equivalent to 1 mm.

**Figure 4 plants-12-01067-f004:**
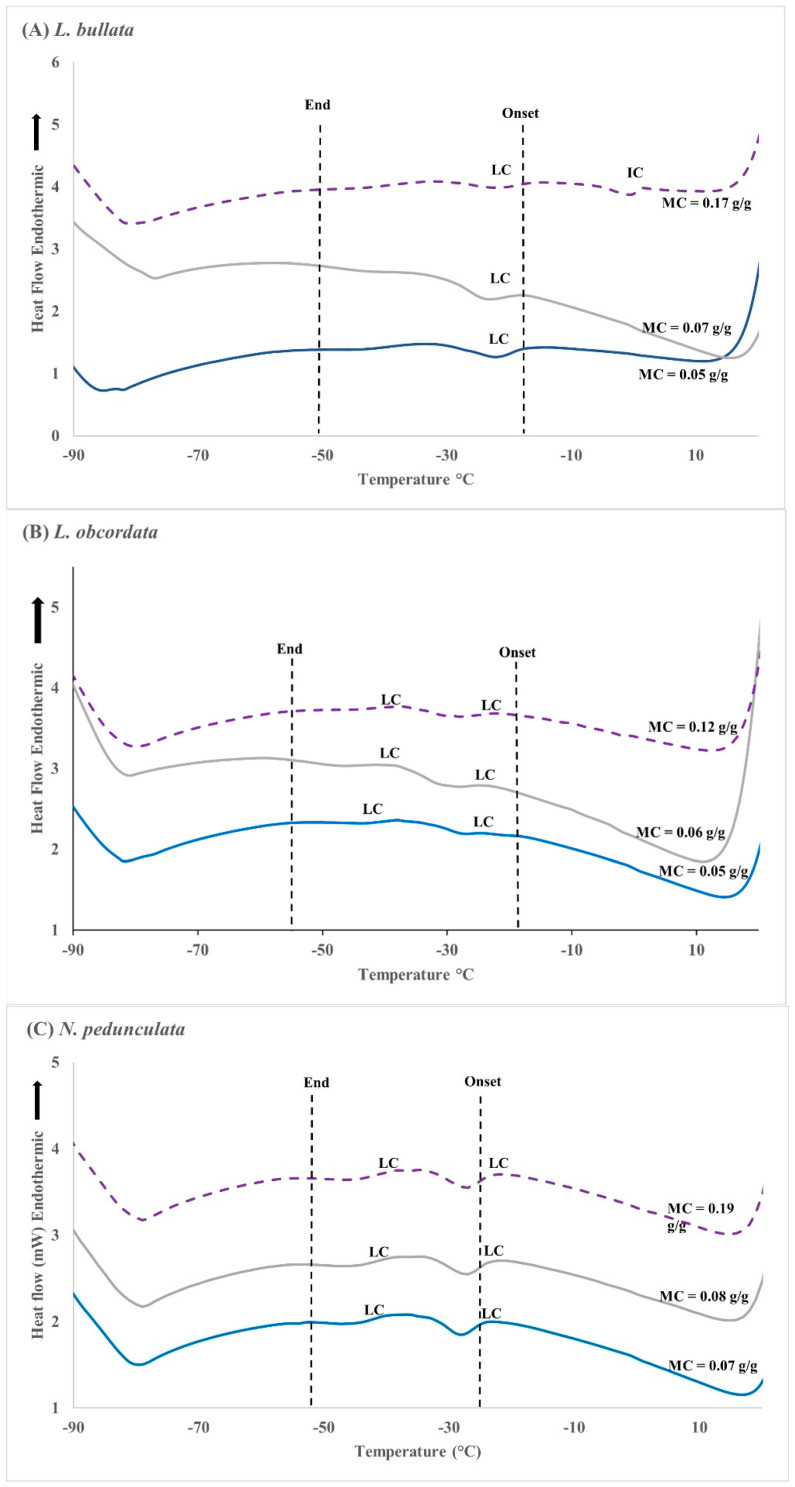
Differential scanning calorimetry (DSC) cooling thermograms for *Lophomyrtus bullata* (**A**), *L. obcordata* (**B**), and *Neomyrtus pedunculata* (**C**) seed equilibrated to 5% (blue line), 15% (grey line), and 30% relative humidity (eRH) (broken purple line). Ice crystallisation (IC), lipid crystallisation (LC), and temperature range from onset to end of crystallisation are indicated. Samples were scanned at 10 min^−1^ from 25 °C to −100 °C. Each sample consisted of three replicates with three seeds in each replicate.

**Figure 5 plants-12-01067-f005:**
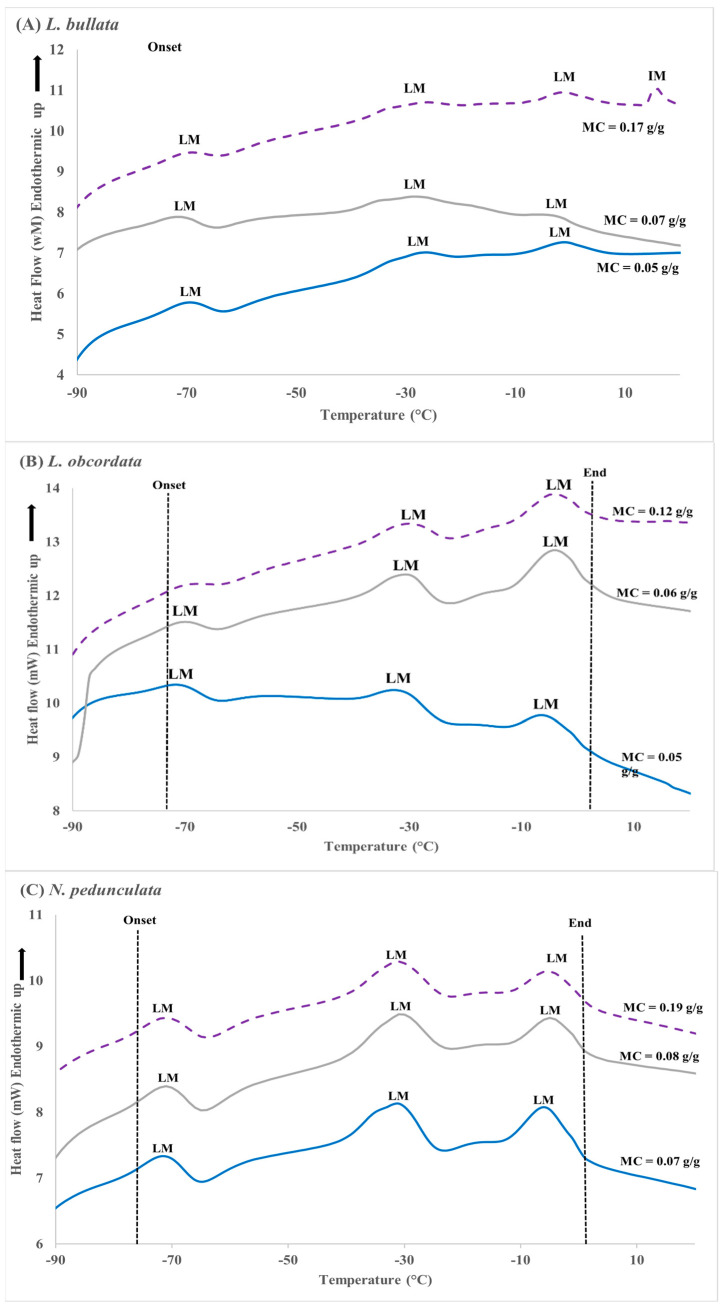
Differential scanning calorimetry (DSC) warming thermograms for *Lophomyrtus bullata* (**A**), *L. obcordata* (**B**), and *Neomyrtus pedunculata* (**C**) seed equilibrated to 5% (blue line), 15% (grey line), and 30% relative humidity (eRH) (broken purple line). Lipid melt transitions are identified as LM and ice melt as IM. Samples were scanned at 10 min^−1^ from −100 °C to 25 °C. Each sample consisted of three replicates with three seeds in each replicate.

**Table 1 plants-12-01067-t001:** Fruit weight, number of seed, and seed characteristics of *Lophomyrtus bullata*, *L. obcordata*, and *Neomyrtus pedunculata*. Letters in each column indicate significant differences at *p* < 0.05 based on Fisher’s LSD test.

Species	Fruit		Seed		
	Weight (g)	Seed Number	Weight (mg)	Diameter (mm)	Moisture Content (g H_2_O/g Dry Weight)
*L. bullata*	0.46 ± 0.07	14.1 ± 4.8 ^a^	9.3 ± 1.0 ^a^	4.06 ± 0.68 ^a^	0.52 ± 0.04 ^a^
*L. obcordata*	0.007 ± 0.002	3.1 ± 4.8 ^b^	7.1 ± 1.0 ^b^	1.47 ± 0.23 ^b^	0.35 ± 0.02 ^b^
*N. pedunculata*	n/d	n/d	3.2 ± 1.0 ^c^	0.77 ± 0.05 ^c^	0.15 ± 0.001 ^c^

**Table 2 plants-12-01067-t002:** Fatty acids as a percentage of total composition of fresh *Lophomyrtus bullata* and *Lophomyrtus obcordata* seeds as well as desiccated (~15% equilibrated relative humidity (eRH)) *L. bullata* seed stored for 15 months at 5 °C and −196 °C.

Lipid	*L. obcordata*	*L. bullata*		
	Fresh Seed (%)	Fresh Seed (%)	Desiccated and Stored for 15 Months at 5 °C (%)	Desiccated and Stored for 15 Months at −196 °C (%)
Caproic (C6:0)	0.22	0.13	0.09	0.12
Undecanoic (C11:0)	0.22	0	0	0
Myristic (C14:0)	0.22	0.13	0	0
Palmitic (C16:0)	6.51	7.71	7.55	7.4
Margaric (C17:0)	0.22	0.13	0.06	0.07
Stearic (C18:0)	2.6	3.27	3.12	3.03
Arachidic (C20:0)	0.43	0.53	0.48	0.43
Heneicosanoic (C21:0)	0.22	0	0	0
Behenic (C22:0)	0.22	0.13	0.07	0.08
Tricosanoic (C23:0)	0.22	0.13	0	0
Lignoceric (C24:0)	0.22	0.13	0.06	0.12
**Total SAFA**	**11.78**	**12.29**	**11.46**	**11.27**
Palmitoleic (C16:1)	0.22	0	0	0
Oleic (C18:1)	11.06	11.63	11.35	10.42
Vaccenic (C18:1)	0.43	0.39	0.38	0.36
Eicosenoic (C20:1)	0.22	0.13	0.06	0.06
**Total MUFA**	**11.9**	**12.1**	**11.8**	**10.8**
Linoleic (C18:2)	75.9	74.9	76.1	77.3
Alpha linolenic (C18:3)	0.7	0.7	0.6	0.6
Eicosadienoic (C20:2)	0.2	0	0	0
**Total PUFA**	**76.8**	**75.6**	**76.8**	**77.9**

SAFA = saturated fatty acids; MUFA = monounsaturated fatty acids; PUFA = polyunsaturated fatty acids.

**Table 3 plants-12-01067-t003:** Radicle emergence, germination, and seedling vigour of fresh seed from *Lophomyrtus bullata*, *L. obcordata*, and *L. bullata x L. obcordata*. Parameter values (radicle emergence, germination, and seedling vigour) within each species followed by the same letter do not differ significantly (Tukey’s HSD; *p* < 0.05; *n* = 10).

Species	Radicle Emergence (%) (Mean ± SD)		Germination (%) (Mean ± SD)		Seedling Vigour			
					Seedling Height (mm) (Mean ± SD)		No. of Leaves (Mean ± SD)	
	NS	DS	NS	DS	NS	DS	NS	DS
*L. bullata*	100 ± 0 ^a^	74 ± 17 ^b^	66.7 ± 5.8 ^a^	45.9 ± 26 ^a^	47.1 ± 11.3 ^a^	45.3 ± 10 ^a^	1.9 ± 1.3 ^a^	1.6 ± 1.3 ^a^
*L. obcordata*	85.0 ± 5.8 ^a^	82.5 ± 9.6 ^a^	85.0 ± 5.8 ^a^	82.5 ± 9.6 ^a^	35.7 ± 11.5 ^a^	34.5 ± 9.5 ^a^	1.9 ± 0.7 ^a^	1.7 ± 0.5 ^a^
*L. bullata x L. obcordata*	81.6 ± 7.1 ^a^	78.2 ± 5.6 ^a^	71.2 ± 6.5 ^a^	66.3 ± 8.6 ^a^	NT	NT	NT	NT

NS = non-stratified; DS = dry stratified; NT = not tested.

**Table 4 plants-12-01067-t004:** Radicle emergence, germination, and seedling vigour for *Neomyrtus pedunculata* seed exposed to wet and dry stratification for 4 and 20 weeks. Parameter values (radicle emergence, germination, and seedling vigour) followed by the same letter do not differ significantly (Tukey’s HSD; *p* < 0.05; *n* = 10).

Stratification Time (Weeks)	Radicle Emergence (%) (Mean ± SD)		Germination (%) (Mean ± SD)		Seedling Vigour			
					Seedling Height (mm) (Mean ± SD)		No. of Leaves (Mean ± SD)	
	WS	DS	WS	DS	WS	DS	WS	DS
4	37.5 ± 13.5 ^a^	0 ^c^	37.5 ± 13.5 ^a^	0 ^c^	15.8 ± 3.6 ^a^	NG	1.1 ± 0.3	NG
20	71.9 ± 21.9 ^b^	0 ^c^	71.9 ± 21.9 ^b^	0 ^c^	20.8 ± 8.5 ^a^	NG	1.1 ± 0.3	NG

DS = dry stratified; WS = wet stratified; NG = no germination.

**Table 5 plants-12-01067-t005:** Desiccation treatment, moisture content, radicle emergence, germination, and seedling vigour (seedling height and number of leaves) for desiccated *Lophomyrtus bullata*, *L. obcordata*, *L. bullata x L. obcordata*, and *N. pedunculata* seed. Fresh seed is represented by non-stratified seed in *L. bullata*, *L. obcordata*, and *L*. *bullata x L. obcordata* with seed stratified wet for 20 weeks representing fresh seed for *N. pedunculata*. Parameter values (radicle emergence, germination, and seedling vigour) for each species followed by the same letter do not differ significantly (Tukey’s HSD; *p* ≤ 0.05; *n* = 10).

Species	Desiccation Treatment	Moisture Content (g/g) (Mean ± SD)	Radicle Emergence (%) (Mean ± SD)	Germination (%) (Mean ± SD)	Seedling Vigour	
					Height (mm) (Mean ± SD)	No. of Leaves (Mean ± SD)
*L. bullata*	Fresh	0.52 ± 0.04	100 ± 0 ^a^	66.7 ± 5.8 ^a^	47.1 ± 11.3 ^a^	1.9 ± 1.3 ^a^
	30% equilibrated relative humidity (eRH)	0.42 ± 0.03	75.0 ± 17.3 ^ab^	70.0 ± 24.5 ^a^	33.2 ± 8.8 ^a^	1.6 ± 0.5 ^a^
	15% eRH	0.095 ± 0.001	47.5 ± 20.6 ^b^	42.5 ± 17.1 ^a^	31.1 ± 8.4 ^a^	1.8 ± 0.5 ^a^
*L. obcordata*	Fresh	0.35 ± 0.02	85 ± 5.8 ^a^	85 ± 5.8 ^a^	37.1 ± 9.5 ^a^	1.9 ± 0.7 ^a^
	30% eRH	0.2 ± 0.004	87.1 ± 6.3 ^a^	87.1 ± 6.3 ^a^	35.7 ± 11.5 ^a^	1.9 ± 0.4 ^a^
	15% eRH	0.094 ± 0.01	85 ± 5.8 ^a^	85 ± 5.8 ^a^	43.1 ± 11.0 ^a^	2.0 ± 0.7 ^a^
*L. bullata x L. obcordata*	Fresh	0.23 ± 0.003	81.6 ± 7.1 ^a^	71.2 ± 6.5 ^a^	NT	NT
	30% eRH	NT	NT	NT	NT	NT
	15% eRH	0.094 ± 0.004	54.6 ± 17 ^b^	54.6 ± 17 ^a^	NT	NT
*N.* *pedunculata*	Fresh	0.15 ± 0.001	71.9 ± 21.9 ^a^	71.9 ± 21.9 ^a^	20.8 ± 8.5 ^a^	1.1 ± 0.3 ^a^
	30% eRH	0.12 ± 0.006	NT	NT	NT	NT
	15% eRH	0.06 ± 0.001	4.0 ± 8.0 ^b^	4.0 ± 8.0 ^b^	10.5 ± 0 ^b^	0 ^b^

NT = not tested.

**Table 6 plants-12-01067-t006:** The combined effects of desiccation (15% and 30% equilibrated relative humidity (eRH)) and temperature exposure (5 °C, −18 °C, and −196 °C) for 14 days on the seed radicle emergence, germination, and seedling vigour of *Lophomyrtus bullata*, *L. obcordata*, *L. bullata x L. obcordata*, and *Neomyrtus pedunculata*. Values followed by the same letter for various parameters (radicle emergence, germination, seedling size, and number of leaves) within each species do not differ significantly (Tukey HSD, *p* ≤ 0.05, *n* = 10).

Species	Desiccation Treatment	Moisture Content (g/g) (Mean ± SD)	Temperature (°C)	Radicle Emergence (%) (Mean ± SD)	Germination (%) (Mean ± SD)	Seedling Vigour	
						Height (mm) (Mean ± SD)	No. of Leaves (Mean ± SD)
*L. bullata*	Fresh	0.52 ± 0.04	NA	100 ± 0 ^a^	66.7 ± 5.8 ^ab^	47.1 ± 11.3 ^a^	1.9 ± 1.3 ^a^
	30% eRH	0.42 ± 0.03	NA	75.0 ± 17.3 ^ab^	70.0 ± 24.5 ^a^	33.2 ± 8.8 ^b^	1.6 ± 0.5 ^ab^
			5	76.7 ± 5.8 ^ab^	60.0 ± 10 ^ab^	33.2 ± 8.8 ^b^	1.8 ± 0.5 ^ab^
			−18	47.5 ± 12.6 ^bc^	35.0 ± 19.1 ^ab^	28.6 ± 7.6 ^bc^	1.3 ± 0.5 ^ab^
			−196	44.4 ± 13.3 ^bc^	31.9 ± 9.0 ^ab^	24.8 ± 10 ^bc^	0.8 ± 0.5 ^ab^
	15% eRH	0.095 ± 0.001	NA	47.5 ± 20.6 ^bc^	42.5 ± 17.1 ^a^	31.1 ± 8.4 ^bc^	1.8 ± 0.5 ^a^
			5	47.6 ± 20.5 ^bc^	41.0 ± 10.1 ^ab^	31.1 ±8.4 ^bc^	1.8 ± 0.5 ^a^
			−18	57.5 ± 6.6 ^bc^	49.1 ± 20.7 ^ab^	26.1 ± 7.3 ^bc^	1.3 ± 0.4 ^ab^
			−196	35.6 ± 9.9 ^c^	23.6 ± 10.5 ^b^	20.8 ± 8.1 ^c^	0.8 ±0.5 ^b^
*L. obcordata*	Fresh	0.35 ± 0.02	NA	85.0 ± 5.8 ^a^	85.0 ± 5.8 ^a^	37.1 ± 9.5 ^a^	1.9 ± 0.7 ^a^
	30% eRH	0.2 ± 0.004	NA	87.1 ± 6.3 ^a^	87.1 ± 6.3 ^a^	35.7 ± 11.5 ^a^	1.9 ± 0.4 ^a^
			5	84.6 ± 12.6 ^a^	78.8 ± 7.4 ^a^	33.7 ± 9.5 ^a^	1.9 ± 0.7 ^a^
			−18	68.8 ± 1.9 ^a^	58.3 ± 0 ^b^	36.7 ± 10.3 ^a^	1.9 ± 0.7 ^a^
			−196	72.2 ± 6.9 ^a^	68.9 ± 1.9 ^ab^	28.7 ± 11.3 ^b^	1.7 ± 0.8 ^b^
	15% eRH	0.094 ± 0.01	NA	85 ± 5.8 ^a^	85 ± 5.8 ^ab^	43.1 ± 11.0 ^a^	2.0 ± 0.7 ^a^
			5	84.6 ± 12.6 ^a^	71.6 ± 15.2 ^ab^	43.0 ± 11.0 ^a^	2.0 ± 0.7 ^a^
			−18	68.6 ± 8.0 ^a^	65 ± 4.9 ^ab^	28.2 ± 9.2 ^b^	1.6 ± 0.7 ^b^
			−196	84.1 ± 11.4 ^a^	73.7 ± 25.6 ^a^	25.6 ± 7.6 ^b^	1.6 ± 0.7 ^b^
*L. bullata x L. obcordata*	Fresh	0.23 ± 0.003	NA	81.6 ± 7.1 ^a^	71.2 ± 6.5 ^a^	NT	NT
	30% eRH		NT	NT	NT	NT	NT
	15% eRH	0.094 ± 0.004	NA	54.6 ± 17 ^b^	54.6 ± 17 ^b^	NT	NT
			5	59.0 ± 12.5 ^b^	50.8 ± 10.5 ^b^	NT	NT
			−18	33.3 ± 2.8 ^b^	29.3 ± 2.8 ^c^	NT	NT
			−196	50.3 ± 15.6 ^b^	44.2 ± 12.3 ^b^	NT	NT
*N. pedunculata*	Fresh	0.15 ± 0.001	NA	71.9 ± 21.9	71.9 ± 21.9	20.8 ± 8.5	1.1 ± 0.3
	30% eRH	0.12 ± 0.006	NT	NT	NT	NT	NT
	15% eRH	0.06 ± 0.001	NA	4.0 ± 8.0	4.0 ± 8.0	10.5 ± 0	0
			−18	0	0	0	0

NA = not applicable; NT = not tested.

**Table 7 plants-12-01067-t007:** Seed storage trial for non-desiccated seed and seed desiccated to 15% equilibrated relative humidity (eRH) for *Lophomyrtus bullata*, *L. obcordata*, and L. *bullata x L. obcordata*. Seed was stored at 5 °C, −18 °C, or −196 °C for 12 to 36 months. Values followed by the same letter for various parameters (radicle emergence, germination, seedling size, and number of leaves) within each species do not differ significantly (Tukey HSD, *p* ≤ 0.05, *n* = 10).

Species	Seed Lot	Desiccation Treatment	Moisture Content (g/g)	Time in Storage (Months)	Storage Temperature (°C)	Radicle Emergence	Germination (%)	Seedling Vigour	
								Seedling Height (mm)	No. of Leaves
*L.* *bullata*	Kap 19	ND	0.52 ± 0.04	24	5	0 ^b^	0 ^b^	n/a	n/a
		15% eRH	0.087 ± 0.02		5	37.8 ± 21.7 ^a^	22.5 ± 17.1 ^a^	25.0 ± 8.4 ^a^	0.6 ± 0.8 ^a^
					−18	3.3 ± 3.8 ^b^	3.3 ± 3.8 ^b^	* 40.2 ± 0 ^b^	* 2.0 ± 0 ^b^
					−196	11.4 ± 13.3 ^b^	0 ^b^	n/a	n/a
	Butchers	15% eRH	0.083 ± 0.03	36	−18	9.2 ± 4.3 ^b^	9.2 ± 4.3 ^b^	20.5 ± 0 ^a^	0.3 ± 0.6 ^a^
	Wrights	15% eRH	0.09 ± 0.02	36	−18	6.9 ± 5.4 ^b^	6.9 ± 5.4 ^b^	* 20.9 ± 0 ^a^	* 1.0 ± 0 ^a^
*L.* *obcordata*	Ōtari 19	ND	0.35 ± 0.02	24	5	0 ^c^	0 ^c^	n/a	n/a
		15% eRH	0.08 ± 0.002		5	92.5 ± 9.6 ^a^	80 ± 14.1 ^a^	30.6 ± 9.7 ^a^	1.1 ± 0.3 ^a^
					−18	6.6 ± 9.4 ^c^	3.3 ± 3.8 ^c^	15.3 ± 5.3 ^b^	* 1.0 ± 0 ^a^
	Matai	15% eRH	0.08 ± 0.002	36	−18	12.3 ± 8.9 ^c^	7 ± 5.8 ^c^	18.0 ± 2.5 ^ab^	* 1.0 ± 0 ^a^
	Old Mill		0.08 ± 0.002	36	−18	64.8 ± 9.8 ^b^	41.4 ± 9.8 ^b^	29.6 ± 9.2 ^a^	1.4 ± 0.6 ^a^
*L. bullata x L. obcordata*	Skyline 20	15% eRH	0.08 ± 0.004	12	5	55.0 ± 17.3 ^a^	42.5 ± 5 ^a^	29.2 ± 14.4 ^a^	1.6 ± 0.5 ^a^
					−18	41.6 ± 5.9 ^a^	19.3 ± 7.1 ^b^	31.7 ± 11.6 ^a^	1.3 ± 0.5 ^a^
					−196	33.3 ± 18.0 ^a^	21.7 ± 6.4 ^b^	22.9 ± 9.5 ^a^	1.2 ± 0.6 ^a^

ND = non-desiccated; NT = not tested. * Based on a single seedling.

## Data Availability

The data presented in this study are available on request from the corresponding author. The data are not publicly available due to the privacy statement in the original project.

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
