# Peer review of "Seed Storage Physiology of Lophomyrtus and Neomyrtus, Two Threatened Myrtaceae Genera Endemic to New Zealand"

_plants, 2023, doi:10.3390/plants12051067_

Round 1

Reviewer 1 Report

In my opinion, the manuscript is very interesting for international readers. The authors have written it nicely and provided valuable information in this manuscript. All my comments will find in the revised version of your manuscript.

In addition, the authors need to modify the methodology and incorporate the new reference in the whole manuscript.

Reviewer 2 Report

Very well-written manuscript with interesting data and results. I have nearly nothing to add, just a few remarks.

English is not my mother tongue, therefore I can't judge it. What I noticed is the writing of commas - in many places, a comma before "and" is missing.

line 38: I would appreciate also the IUCN category (if evaluated);

line 265: on picture A, there should be the same line (broken purple) as on pictures B and C;

line 274: typo (grey and not great line, I suppose);

line 307: tomatillos and not tomatoes (Physalis is not a tomato);

line 390: I would recommend starting the sentence: "The DSC thermograms in our study..." - as the previous sentence starts "This study..." and it is quite confusing about what you write;

line 432: the seed weight is usually determined as a weight of 1000 seeds with 8 repetitions, but I understand that you may lack enough material for such a test.

Thank you, I really enjoyed its reading!

Reviewer 3 Report

General recommendations and questions

If the full Latin name for the species has already been given once, the abbreviated form can be used further in the text.

Why is the letter “c.” used before numerical values, for example MC c. 0.35 g/g? This is not explained anywhere.

Tables

Table 1.

What are Damage markers? For seeds? May be Damage markers are seed properties as mentioned in the table title?

Table 2.

Table 2 is somewhat difficult to understand due to the design of the table. As far as I understand - the 1st colon attributes to L. obcordata, the 2nd to L. bullata? Perhaps the two species can be separated more clearly? This is just a suggestion.

Table 3

Seedling vigour as parameter sounds strange. Parameters are plant height, number of leaves. It is recommended to denote the number of samples as n=xxx

Why did you use # instead of No to indicate the number of leaves?

“% mean ± SD”? – I recommended: mean ± SD, %

These notes also apply to other tables.

Table 6.

What is the TZ stain in the table name? What is the “size” in Seedling Vigour’s colon? Is the seedling size the same as the seedling height in the previous tables? Same question about Table 7.

Introduction

Line 53. “Of these variables, seed lipid composition, particularly the thermal stability of the lipids at the storage temperature, is a key predictor of intermediate seed storage physiology that generally contain high oil content [13, 15-18] “. Please write more clearly. What is high in oil?

Results

Line 134. “Seedling vigour was not recorded in the natural hybrid, L. bullata x L. obcordata”. I would recommend avoiding the expression "Seedling vigour was not recorded". Maybe - the parameters characterizing the vigour of the seedlings were not recorded.

Discussion

I would recommend citing previous research to indicate the authors in such cases as:

For example:

311 line. ….as suggested by [16]. More correct would be …. as suggested by Mira et al. [16].

326 line. “For example, [39] found that seed dormancy…..” more correct would be  - Andersson and Milberg [39] found that seed dormancy…..

In general, the article is interesting, contains new knowledge about various aspects of seed physiology, clearly understandable, with rich data material, contains an extensive list of scientific literature. I recommend accepting this article in Plants after minor revision.

Round 2

Reviewer 1 Report

he authors, I am grateful for your hard work in reviewing the manuscript, thus improving its overall quality. Most of the comments have been addressed.